# Towards a High-Affinity Peptidomimetic Targeting Proliferating Cell Nuclear Antigen from *Aspergillus fumigatus*

**DOI:** 10.3390/jof9111098

**Published:** 2023-11-10

**Authors:** Bethiney C. Vandborg, Aimee J. Horsfall, Jordan L. Pederick, Andrew D. Abell, John B. Bruning

**Affiliations:** 1Institute of Photonics and Advanced Sensing (IPAS), The University of Adelaide, Adelaide 5005, Australia; bethiney.vandborg@adelaide.edu.au (B.C.V.); jordan.pederick@adelaide.edu.au (J.L.P.); andrew.abell@adelaide.edu.au (A.D.A.); 2School of Biological Sciences, The University of Adelaide, Adelaide 5005, Australia; 3ARC Centre of Excellence for Nanoscale BioPhotonics, The University of Adelaide, Adelaide 5005, Australia

**Keywords:** PCNA, DNA replication proteins, non-tag purification, peptide characterization, *Aspergillus fumigatus*, X-ray crystallography

## Abstract

Invasive fungal infections (IFIs) are prevalent in immunocompromised patients. Due to alarming levels of increasing resistance in clinical settings, new drugs targeting the major fungal pathogen *Aspergillus fumigatus* are required. Attractive drug targets are those involved in essential processes like DNA replication, such as proliferating cell nuclear antigens (PCNAs). PCNA has been previously studied in cancer research and presents a viable target for antifungals. Human PCNA interacts with the p21 protein, outcompeting binding proteins to halt DNA replication. The affinity of p21 for hPCNA has been shown to outcompete other associating proteins, presenting an attractive scaffold for peptidomimetic design. p21 has no *A. fumigatus* homolog to our knowledge, yet our group has previously demonstrated that human p21 can interact with *A. fumigatus* PCNA (afumPCNA). This suggests that a p21-based inhibitor could be designed to outcompete the native binding partners of afumPCNA to inhibit fungal growth. Here, we present an investigation of extensive structure–activity relationships between designed p21-based peptides and afumPCNA and the first crystal structure of a p21 peptide bound to afumPCNA, demonstrating that the *A. fumigatus* replication model uses a PIP-box sequence as the method for binding to afumPCNA. These results inform the new optimized secondary structure design of a potential peptidomimetic inhibitor of afumPCNA.

## 1. Introduction

Invasive fungal infections (IFIs) are a prevalent cause of death in immunocompromised patients [1]. A major fungal pathogen causing such infections is *Aspergillus fumigatus*, a filamentous fungus that is usually present in decaying organic matter [2,3]. The conidia of *A. fumigatus* inhaled from the environment can be cleared from the lungs by a healthy immune system. However, when this fails, the conidia can reach the lower airways and evade host immune cells [4]. This can result in the infection of the bronchi and sinuses and dissemination to the brain and other vital organs through the circulatory system [5,6]. This is known as invasive aspergillosis. When invasive aspergillosis invades the nervous system, it has mortality rates of up to 90% [7].

As the infection rates of *A. fumigatus* increase, more species and therefore differing antifungal resistances arise, which have been associated with negative clinical outcomes [8]. Although many antifungals are available, the mortality rates remain high due to the development of drug resistance in *A. fumigatus* [9]. Current treatments of invasive aspergillosis target the components of the cell membrane: for example, the antifungal amphotericin B [10]. Unfortunately, in addition to increasing resistance rates, amphotericin B is associated with severe side effects, the most notable being kidney and liver toxicity [11]. There has been increasing resistance to the antifungal triazole reported via unknown resistance mutations [12]. As such, there is an urgent need for improved diagnostic protocols and a broader range of antifungal options [13].

The DNA replication process is a desirable target for an antifungal product; hence, *A. fumigatus* proliferating cell nuclear antigen (afumPCNA) has been hypothesised as a fungal target for the development of new antifungals [14]. The PCNA processivity factor is also known as the sliding clamp, as DNA slides through its central cavity. PCNA functions as a docking platform to allow DNA polymerases and a host of DNA replication and repair machinery to interact at the replication fork [15]. PCNA has been proposed as a target for the treatment of multiple diseases as it is essential for cell replication, and its absence has been shown to cause embryonic lethality [16,17]. 

The X-ray crystal structure of apo afumPCNA solved at 2.6 Å resolution [14] shows a trimeric tertiary structure that is similar to that of hPCNA by the superimposition of the structures (PDB: 8GJF and PDB: 7KQ1), this revealed a root-square standard deviation (RMSD) of 0.939 Å. Despite this, the proteins only have a sequence similarity of 53% [14]. The PCNA homotrimer comprises two domains, with each containing two alpha helices and nine beta strands (Figure 1b), connected by a motif known as the interdomain connecting loop (IDCL), which forms part of the PCNA surface with which binding domain proteins interact. The negatively charged beta sheets allow interactions with replication and repair proteins, and the positively charged alpha helices allow non-specific interactions with double-stranded DNA on the inside of the sliding clamp (Figure 1a). It has been hypothesised that the difference in amino acid sequences present at the surface of PCNA, to which interacting partners bind, could allow for the specificity of afumPCNA over hPCNA in the design of a fungal replication inhibitor [14]. This illustrates the importance of investigating the structure of afumPCNA to understand how interacting peptides bind in aiding rational drug design.

In an effort to target PCNA, the most thoroughly characterised peptide inhibitor is derived from the tumour suppressor protein p21. Protein p21 (also known as cyclin-dependent kinase inhibitor 1), a cyclin-dependent kinase inhibitor, binds hPCNA to outcompete binding partners in order to halt DNA replication for repair systems [20], thus regulating the cell cycle during DNA damage. The affinity of peptides derived from p21 binding to hPCNA has been shown to be much higher than other associating proteins [18,21]. PCNA-interacting proteins, including p21, are allowed access to DNA by interacting with PCNA via the PCNA-interacting protein (PIP) box sequence, Q_144_X_145_X_146_Φ_147_X_148_X_149_Ψ_150_Ψ_151_, a consensus sequence in which the glutamine residue (Gln_144_) binds the ‘Q pocket’ of the PCNA hydrophobic patch, Φ_147_ represents a hydrophobic residue, and Ψ_150_ and Ψ_151_ represent aromatic residues. The sequence and affinity of the PIP-box are theorised to correlate with the protein’s function [22]. The combination of the hydrophobic residue and two aromatic residues forms a hydrophobic plug that inserts into the PCNA surface and twists the peptide’s backbone from residues 147 to 151 into a 3_10_ helix that is conserved between binding partners. This secondary structure is critical for high-affinity binding. Other interactions that increase affinity are the ionic charged interactions of the C-terminal flank of the PIP-box with the surface of PCNA, and the N-terminal flank of the PIP-box creates an antiparallel β-sheet with the C-terminus of PCNA. 

The fluorescence polarisation experiments of afumPCNA and a 22 amino acid peptide derived from the C-terminus of (human) p21 containing the PIP-box (139–160) have demonstrated their interaction, suggesting that afumPCNA interacts with DNA binding proteins using a similar PIP-box mechanism compared to the human system [14]. Given that *Aspergillus fumigatus* does not have a known p21 equivalent and this p21-derived peptide shows high-affinity interactions, further investigation into human PIP-box sequence interactions with afumPCNA may indicate the characteristics of a high-affinity mimetic. The p21 scaffold may serve as a useful starting point for designing the peptide inhibitors of afumPCNA. This also suggests that an artificial PIP-box could be designed specifically to disrupt the function of afumPCNA and highlights afumPCNA as a potential drug target for treating fungal infections. Fungal PCNA-interacting proteins were investigated as a means of probing these unknown PIP-box sequences; therefore, they can uncover interactions that could be advantageous to a mimetic. These PIP-box candidates were investigated in fungal proteins DNA polymerase (DNAPol), DNA ligase (DNALig), flap endonuclease 1 (FEN1), and replication factor C (RFC).

Here, we present the first structure of a PIP-box peptide bound to afumPCNA and characterise the interactions of afumPCNA with predicted fungal protein PIP-box candidates. Via binding affinity assays and X-ray crystallography studies, the findings support the hypothesis that the fungal replication model uses a PIP-box sequence as the method for binding to fungal PCNA, and a rational design for a potential peptidomimetic is presented.

## 2. Materials and Methods

### 2.1. Peptides

The following peptides were obtained and synthesised by Genscript Biotech, Singapore at a purity of >95%, and they were purified via HPLC. Peptides denoted with * were designed by *B. Vandborg*. The sequences are shown in bold.

5FAM-p21-22mer **(5FAM)-GRKRRQTSMTDFYHSKRRLIFS**

p21µ p21µ-15mer **KRRQTSMTDFYHSKR**

p21µ-afumDNALIG **KRRQRVRSIASFFHSKR***

p21µ-afumDNAPOL **KRRQKELSRFDFHSK***

p21µ-afumFEN1 **KRRQSRLEGFFHSKR***

p21µ-afumRFC **KRRMPTDIRNFFHSKR***

The following peptides were synthesised by Fmoc SPPS, as described below in Section 2.1.1; each has a C terminal carboxyl amide. The sequences are shown in bold.

p21µ (p21µ-15mer) **KRRQTSMTDFYHSKR**

p21µ-RD2 **KRRQTRITEYFHSKR**

p21µ-Q144M **KRRMTSMTDFYHSKR**

p21µ-T145K **KRRQKSMTDFYHSKR**

p21µ-T145D **KRRQDSMTDFYHSKR**

p21µ-S146R **KRRQTRMTDFYHSKR**

p21µ-M147L **KRRQTSLTDFYHSKR**

p21µ-M147I **KRRQTSITDFYHSKR**

p21µ-D149E **KRRQTSMTEFYHSKR**

p21µ-F150Y **KRRQTSMTDYYHSKR**

p21µ-Y151F **KRRQTSMTDFFHSKR**

p21µ-FY150151YF **KRRQTSMTDYFHSKR**

#### 2.1.1. Peptide Synthesis by Fmoc SPPS

All peptides were prepared on Rink Amide functionalized polystyrene resin (Agilent, Santa Clara, CA, USA) and synthesized via Fmoc/*t*Bu solid-phase peptide synthesis (SPPS), as previously described [18]. The peptides were purified via semi-preparatory RP-HPLC, and the purity and identity were confirmed via analytical HPLC and MS, as previously reported [18].

### 2.2. Expression of Recombinant afumPCNA

*A. fumigatus* PCNA was expressed as described in Vandborg 2023 [23].

A glycerol stock of *E. coli* BL21 (DE3) cells carrying a codon-optimized afumPCNA-pMCSG19 plasmid was grown in a 100 mL overnight culture. Two 1 L baffled flasks containing LB with 100 μg/mL of ampicillin were inoculated with 50 mL of the overnight culture. Cultures were incubated at 37 °C until OD600 = 0.7, and expression was induced with a final concentration of 0.5 mM IPTG. Cultures were grown overnight at 16 °C and shaking occurred at 200 rpm. Cultures were pelleted at 5000× *g* for 20 min. After removing the supernatant, pellets were resuspended in 20 mL 20 mM Tris-HCl pH 7.5, 20 mM NaCl, and 2 mM DTT and then lysed via sonication at 70% amplification for 20 s with a 40 s waiting period for 25 cycles. Lysate was clarified via pelleting at 45,000× *g* for 45 min, and the supernatant was collected for purification.

### 2.3. Purification of Recombinant afumPCNA 

*A. fumigatus* PCNA was purified as described in Vandborg 2023 [23].

Buffer solutions were filtered before being used. Clarified lysate containing afumPCNA was first purified at 4 °C via fast protein liquid chromatography (FPLC) using anion exchange chromatography and two DEAE columns in series (HiTrap DEAE FF 5 mL column). They were then quilibrated in Buffer A (20 mM Tris-HCl pH 7.5, 20 mM NaCl, 2 mM DTT), and afumPCNA was eluted using a linear gradient (0.02 M–0.7 M NaCl). Fractions containing afumPCNA were pooled, and ammonium sulphate was added dropwise to a final concentration of 1.5 M from a stock solution of 3 M ammonium sulphate. The sample was allowed to stir gently for 1 h at 4 °C to allow DNA–protein dissociation, and then it was applied to hydrophobic interaction chromatography (HiTrap Phenyl FF (high sub) 5 mL column) and equilibrated in Buffer B (20 mM Tris-HCl pH 7.5, 20 mM NaCl, 2 mM DTT, 0.5 mM EDTA, 1.5 M ammonium sulphate and eluted in Buffer C (20 mM Tris-HCl pH 7.5, 2 mM DTT, and 0.5 mM EDTA) with a reverse linear gradient (1.5 M–0 M ammonium sulphate). Fractions containing afumPCNA were pooled and dialyzed overnight in Buffer A. afumPCNA was then applied to a second anion exchange step. The Q Sepharose column (5 mL Q Sepharose FF column (GE)) was equilibrated in Buffer A, and the protein was eluted using a linear gradient (0.02 M–0.7 M NaCl). Fractions containing afumPCNA were pooled and dialyzed overnight in 20 mM Tris-HCl pH 7.5, 10% *v/v* glycerol, 2 mM DTT, and 0.5 mM EDTA. The protein for crystallography was concentrated to ∼10 mg/mL using a centrifugal filter unit (50 kDa molecular mass cut off) and stored at −80 °C.

### 2.4. Surface Plasmon Resonance Protocol

Surface plasmon resonance (SPR) experiments were performed as previously described [18]. The running buffer used for ligand attachment and analyte-binding experiments was a 10 mM HEPES buffer with 150 mM NaCl, 3 mM EDTA, and 0.05% Tween-20, adjusted to pH 7.4 with 2 M NaOH. A GE CM5 (series S) sensor chip was primed with the running buffer and preconditioned with successive injections (2 × 50 s, 30 μL/min) of 50 mM NaOH, 10 mM HCl, 0.1% SDS, 0.85% H_3_PO_4_, and 50 mM glycine pH 9.5, respectively. The surface was then activated with an injection of 0.2 M 1-ethyl-3-(3-dimethylaminopropyl)carbodiimide (EDC) and 50 mM N-hydroxysuccinimide (NHS) (600 s, 10 μL/min). *A. fumigatus* PCNA (5 μL, 12 mg/mL) was diluted into the running buffer (245 μL). Upon the preactivation of the surface, the protein was further diluted to a final concentration of 25 μg/mL in 10 mM NaAc (~pH 4.6). This solution was immediately injected over the target flow cell (10 μL/min) to immobilize ~1500 RU. Both the target and reference flow cells were then blocked with 1.0 M ethanolamine at pH 8.5 (600 s, 10 μL/min). The chip was left to stabilize before sample injections commenced.

Peptide stock solutions for use in SPR experiments were prepared in MilliQ water. The peptide stock concentration was determined via 205 nm absorbance with NanoDrop2000. The ε_205_ for each peptide was calculated using an online calculator (http://nickanthis.com/tools/a205.html, accessed on 22 August 2022 [24]); however, additional glycine residue was added to each peptide sequence to account for the terminal amide of the peptides synthesized in-house. The peptide stock solution’s concentration was then calculated using Beer’s Law.

Steady-state affinity experiments were conducted at a flow rate of 30 μL/min, with a starting contact time of 40 s and dissociation of 30 s. A 2-fold serial dilution was performed for each peptide, with 8 samples injected sequentially from the lowest to highest concentrations; they were preceded by a buffer-only blank injection. After each injection, the surface was regenerated with 2 M NaCl (2 × 30 s, 25 μL/min). All data were analyzed using the GE Biosystems Biacore S200 Evaluation Software, Version 1.0 (Build: 20). All data are summarized in Table 1.

### 2.5. Protein-Peptide Co-Crystallization Experiments

To form the protein–peptide complex, afumPCNA was mixed with the peptide of interest at a molar ratio of 1:1.2. After incubation on ice for 30 min, the sample was pelleted at 16,000× *g* for 10 min to remove aggregates. Crystals were grown via the hanging drop vapor diffusion method in 24-well plates containing 500 μL of well solution by mixing 1 μL of protein and peptide with equal volumes of well solutions. The diffracting crystals of afumPCNA bound to p21µ grew in 0.2 M Tacsimate pH 4.0 0.1M Na Acetate and 16% PEG 3350 (Hampton Research Aliso Viejo, CA, USA, product code HR2-591) at 16 °C after 3 weeks (Appendix A). The diffracting crystals of afumPCNA bound with p21µ-afumRFC grew in 0.2M Tacsimate pH 4.0 0.1M Na Acetate and 16% PEG 3350 (HR2-591) in a tray at 16 °C after 3 weeks (Appendix A). Crystals were mounted on cryo-loops, and they were cryoprotected using paratone-N and flash-cooled in liquid nitrogen. Data were collected at 100 K using the MX1 beamline at the Australian Synchrotron (Clayton, VIC, Australia). Diffraction data were indexed and integrated using XDS (X-ray Detector Software), Version January 10, 2022 [25]. Pointless (CCP4i) [26] was used to create an mtz reflection file for scaling. Data were scaled and merged using Aimless (CCP4i) [27] at a resolution of 2.0 Å for afumPCNA bound with p21µ and 2.30 Å for afumPCNA bound with p21μafumRFC. The phase problem was solved via molecular replacements using Phaser MR (CCP4i) [28] and a search model (PDB: 5TUP). Solutions were refined in Phenix Refine [29,30] in iterative rounds with manual rebuilding in Coot [31] (Appendix A). Data collection and refinement statistics are summarized in Appendix A. The final structures of afumPCNA bound with p21µ and p21µRFC are deposited on the RCSB database under accession numbers 8GJF and 8GJ5, respectively.

### 2.6. Computational Modelling

The models of peptides bound to afumPCNA were constructed using the structure of afumPCNA bound with the p21µ peptide (PDB: 8GJF) as a starting template, and the necessary, deleted, and unresolved side chains of residues were modelled into the computational structure. 

The manual refinement of the computational linker was carried out in Coot [31]. Energy minimisation/annealing (n = 30) for refinement was carried out in ICM-Pro Molsoft [32,33]. Refined models were analysed using PyMOL Version 1.2 [19] to validate the model by comparing it against p21µ (PDB: 8GJF) and assessing side chain interactions. The resulting structures were visualized in PyMOL [19], and they are depicted in Appendix A.

## 3. Results

### 3.1. A p21 Peptide Library Interacts with afumPCNA in a Similar Trend Compared to hPCNA

The p21-derived peptide (139–160) (Table 1) [21] was previously shown to bind to afumPCNA via fluorescence polarization with a K_D_ of 3.1 µM [14]. Here, we build on this observation and interrogate the binding of afumPCNA with respect to various peptides. Previously, a shorter scaffold of this 22 amino acid p21 peptide was derived and synthesized: p21µ (141–155). It is 14 amino acids long, and it retained high-affinity binding, which was used to construct a library with variations relative to PIP-box residues [18] as a rational starting point for the investigation of the fungal binding site.

The p21 PIP-box contains a conserved glutamine residue that binds a conserved hydrophobic pocket on hPCNA; this was shown to be valuable in binding afumPCNA. The p21 glutamine residue (Gln_144_), which binds the ‘Q pocket’ of the hydrophobic patch on human PCNA, forms two hydrogen bonds relative to the carbonyl backbone moieties of residues Ala_252_ and Pro_253_ of the PCNA main chain. These residues are conserved in the afumPCNA sequence (Figure 2); hence, as for the hPCNA investigation, the modification of Gln_144_ into Met, as in p21μ-Q144M, reduces the binding of peptide p21μ-Q144M to afumPCNA from 265.1 nM to 41,400 nM. 

The importance of residues in the non-conserved position of the PIP-box was previously shown to be important in binding to the PCNA surface in the human system [34]. To probe the effect of altering amino acids in this region of the PIP-box and its affinity to afumPCNA, the peptide was altered from Ser_146_ to an Arg residue. This produced an affinity of K_D_ 64.4 nM, improved from the p21μ binding affinity of K_D_ 265.1 nM. This was also previously observed in hPCNA, which was hypothesised to be due to an increase in side chain length [18]. The Ser to Arg variation changes the distance between residues, strengthening the intramolecular hydrogen bond to the carbonyl of Asp_149_, stabilising the peptide’s 3_10_ helical structure (Appendix A). This suggests that lengthening the side chain would also improve the binding of the p21μ-D149E peptide to afumPCNA; however, an Asp_149_ modification to Glu showed reduced binding affinity with a K_D_ of 400.6 nM, and this was possibly due to the negatively charged side chain having an unfavourable interaction with the binding surface of afumPCNA (Appendix A). 

The Φ_147_ of the PIP-box consensus sequence is conserved amongst most hPCNA binding proteins as methionine residue. The methionine side chain binds to a hydrophobic pocket under the IDCL. Isoleucine is also observed in this position as it is a suitable replacement for methionine. The modification of Met_147_ to Ile produced affinity for the afumPCNA binding site of K_D_ 37 nM and a 7-fold improvement in affinity for afumPCNA compared to p21μ, a trend observed in the previous hPCNA investigation [18] (Appendix A). 

The two aromatic residues at positions 7 and 8 of the p21 PIP-box each form a hydrophobic plug that inserts their side chains into the hydrophobic patches on the hPCNA surface, which helps form the PIP-box’s peptide backbone into a 3_10_ helix. The Phe_150_ to Tyr modification has a K_D_ affinity of 75 nM for afumPCNA (Appendix A), and Tyr_151_ to Phe has a K_D_ of 167 nM (Appendix A); both improved from p21µ (265.1 nM). This result is similar to that found in hPCNA. Combining these advantageous modifications in the p21μ-FY150151YF peptide gives a K_D_ affinity of 96.4 nM for afumPCNA (Appendix A). 

### 3.2. Fungal Protein PIP-Box Candidate Discovery Displays a Surprising afumPCNA Interaction

The proteins important to DNA replication, which were hypothesised to interact with the sliding clamps via a PIP-box binding motif, were selected for investigation. These include afum DNA polymerase (DNAPol), DNA ligase (DNALig), flap endonuclease 1 (FEN1), and replication factor C (RFC). Candidates for PIP-box sequences were chosen based on their location in the fungal protein sequence. Sequence candidates were those that fit the model of approximately an eight-residue section, beginning with Gln (in all cases except the RFC) and hydrophobic residues and ending with aromatic residues (Table 2). These candidate PIP-boxes were used in the p21μ peptide scaffold and PIP-box flanking regions to create fungal origin peptides for affinity and structural characterization, as the p21μ peptide scaffold provides a functional starting mechanism and the flanking regions have been shown to be important for interacting with the IDCL.

The major differences between human and fungal PIP-boxes appear at the non-conserved residues of the canonical PIP-box sequence. In particular, in the p21μafumDNALig and p21μafumRFC sequence, additional residues were interspaced with conserved residues, possibly interfering with the alignment of the canonical structure and the contact with the protein’s surface. To elaborate, the RFC1 PIP-box found in humans has the correct amount of non-conserved residues, but the candidate for *A. fumigatus* has two extra residues between the conserved methionine residue and the conserved hydrophobic residue isoleucine (Table 2). 

Each *A. fumigatus* PIP-box sequence exhibits a Gln_144_-conserved residue, except the RFC sequence. This significant difference between the human and fungal candidate PIP-box leads to the hypothesis that the p21μafumRFC peptide could not bind to afumPCNA with high affinity. However, it is surprisingly bound with <100 nM affinity. 

### 3.3. X-ray Crystallography Study of the p21µ Peptide Bound to afumPCNA

The first co-crystal structure of afumPCNA bound with the p21µ scaffold peptide was solved at 2.0 Å resolution (PDB: 8GJF) in order to examine the details of the binding interaction (Figure 3). The structure shows that the overall fold of the p21µ peptide bound to the surface of afumPCNA is similar to the structure of the p21µ peptide bound to hPCNA (PDB: 7KQ1), as illustrated by the RMSD value of 0.939 Å. The p21µ peptide in the afumPCNA structure (PDB: 8GJF) displays a notable charged interaction between Arg_143_ and Glu_149_, a 3.2 Å salt bridge interaction (Figure 4b) that was not previously shown as the extended Arg_143_ side chain was not present in the hPCNA crystal structure (Figure 4a), illuminating a new interaction that also strengthens the 3_10_ helical structure. 

Differences in the protein sequence of afumPCNA and hPCNA account for the shift in affinity with the conservation of the secondary structure of the peptide. There is structural conservation around the 3_10_ helical secondary structure and PIP-box, with more variability on the N- and C-terminus (Figure 4c) likely due to the mobility of the ends of the peptide and changes in IDCL residues. Previous literature interpreting the difference in the binding of p21-based peptides to afumPCNA compared to hPCNA used molecular dynamics to illustrate that the weakness of the p21 peptide (139–160) because afumPCNA came from differences in these protein binding domain residues [14]. One prominent example is residue His125, which forms an antiparallel β-sheet with the C-terminal residues of p21 peptide on hPCNA (139–160); however, in the afumPCNA structure, afumPCNA His125 obstructs the formation of a favorable side chain hydrogen bond with the His152 side chain of the p21 peptide (139–160). This has the effect of pushing the C-terminus to be quite distant from the protein’s surface while not forming the hydrogen bonds of the β-sheet as observed in the human structure. There is also the loss of 3.4 Å hydrogen bond interactions between hPCNA Gln131 and the Tyr151 phenol of p21µ in afumPCNA as this residue is Thr131. 

### 3.4. X-ray Crystallography Study of afumPCNA and p21µ-afumRFC Reveals a Ring-like Structure

The co-crystal structure of afumPCNA bound with the p21µ-afumRFC peptide was solved at a resolution of 2.30 Å in order to examine the structural features of an Afum-derived PIP-box (Figure 5a,b).

The KRRMP amino acids of the p21µ-afumRFC peptide (Table 3) fold over the PIP-box, not interacting with the afumPCNA’s surface (Figure 5c). This is caused by a change in the sequence of the PIP-box compared to the human RFC sequence (Table 3); the inclusion of a proline residue causes a kink, and the backbone carbonyl of ProXXX interacts with the Asp_147_ backbone amide and Asn_150_ residue side chain to stabilize the turn in the peptide (Figure 4c). The second arginine, Arg_143_, is located close to the third arginine, Arg_149_, producing a loop structure (143–149). The Met_144_ side chain does not interact with the afumPCNA surface pocket as Gln_144_ in the p21µ PIP-box does with hPCNA. The Met_144_ backbone amide does interact with the Asn_150_ side chain and in turn also interacts with the Asp_147_ side chain, supporting the 3_10_ helix via extra contacts that hold the compact structure (Figure 4c). 

## 4. Discussion

### 4.1. The p21 Peptide Library Interacts with afumPCNA with Similar Affinity and Structural Trends as hPCNA

The co-crystal structure of afumPCNA bound to the p21µ peptide supports the hypothesis that *A. fumigatus* adopts a PIP-box sequence as a method for proteins to interact with afumPCNA. 

The SPR (Table 1) results indicate that the binding of the p21 peptide library to afumPCNA generally follows the same trends seen in the hPCNA investigation [18]. This includes modifications at similar positions that cause similar changes in binding affinity across the two PCNA species. 

It was previously hypothesized from results in molecular dynamics studies [14] that differences in the residues of the IDCL of afumPCNA and hPCNA could provide specificity for afumPCNA over hPCNA for a peptidomimetic inhibitor. However, this does not seem to be supported, as high-affinity peptidomimetic residues have a similar trend in affinity for hPCNA and afumPCNA. This is directly shown in the binding of rational design mutant 2 (RD2) to afumPCNA (Appendix A) [18], which was specifically designed for hPCNA, but it binds to the surface of afumPCNA with the same structure and a K_D_ of 20.3 nM (Table 1). The two PCNA species cannot be differentiated in specificity through the changes in PIP-box residues via the modifications investigated here.

### 4.2. Fungal Protein Replication Factor C PIP-Box Candidate Pepide Has a High Affinity for afumPCNA

Previously, a p21 peptide with the human RFC PIP-box, p21μ–RFC, which has a seven-amino-acid PIP-box, MDIRKFF, was investigated to understand variations in the canonical sequence, and it was found to have a K_D_ value of 145 nM [18]. It was hypothesised here that this affinity was due to the position of residues Ile_147_, Phe_150_, and Phe_151_, which form a hydrophobic three-pronged plug that inserts into the hydrophobic cleft of hPCNA [18]. Via computational modelling, it was observed that this would result in the extension of the arginine residue of position four over the conserved glutamine pocket in order to interact with hPCNA residue Val_45_ [18].

The p21µ-afumRFC PIP-box has an affinity for afumPCNA of less than 100 nM. This may be accounted for solely by the IIe_147_, Phe_150,_ and Tyr_151_ residues (Table 1), as previous research has shown these conserved residues to be highly favourable, especially IIe_147_. The p21µ-afumRFC PIP-box has a lower affinity for hPCNA than afumPCNA (Table 3). This is not only attributed to the lack of Gln_144_, similarly to that of the human RFC peptide, but also the extra residues (Pro_145_, Thr_146_, and Asn_150_) of the PIP-box for which the canonical positions do not exactly fit the conserved motif. It was hypothesised [18,35] that the Gln_144_ residue was essential to the p21 peptide with respect to high-affinity binding; hence, it is present in p21µ-RD2. The Gln_144_ of p21 is known to contribute significantly to the binding affinity of hPCNA, as a Gln144Ala modification was not able to effectively inhibit DNA replication in vitro [18,35]. Gln_144_ was considered at first to remain important in the Afum binding since the modification of p21μ-Q144M decreases the binding affinity of the p21μ-Q144M peptide to afumPCNA from 265.1 nM to 41.4 µM. This is solely attributed to the single residue change as the secondary structure is maintained (Appendix A). However, here, its importance has still been questioned for the afumPCNA binding domain due to its absence in the candidate fungal RFC PIP-box. The attributes of p21µ-afumRFC affinity for afumPCNA, although the canonical PIP-box is not followed, appear to be the unique secondary structure that is formed, which is discussed further.

### 4.3. The p21μafumRFC Peptide Has a Unique Structure That Could Be Exploited for an Antifungal Treatment

Peptidomimetic drug pipelines often reach the point of requiring a cell-permeable mechanism; a convenient method of improving cell uptake is via the cyclisation of the peptide. Cyclic peptides have been shown to enter the mammalian cell cytosol via multiple mechanisms, including passive diffusion, which is facilitated predominantly by hydrophobic side chains and small amino acid size (approximately 10 amino acids long), and endocytic uptake and endosomal escape [36].

In a structure such as the p21 PIP-box, which creates a 3_10_ helix, constraining this structure via cyclisation would allow the preorganization of the backbone and reduce the entropic cost of forming the secondary structure upon binding. Another advantage is that cyclical peptides may have improved cell permeability, which has been investigated in previous studies [37,38]. The investigation studied such macrocycles bound hPCNA with K_D_ values ranging from 570 nM to 3.86 μM, with a bimane-constrained peptide proving to be the most potent. This peptide was also cell-permeable and localized to the cell cytosol of breast cancer cells (MDA-MB-468). The 3_10_-helical structure was present in the computationally modelled structure. However, the analysis showed the peptide did not have a rigid 3_10_ helix in the solution when not bound to PCNA as NMR revealed it was not present in the solution [38]. This suggested that the pre-defining of a peptide backbone may not improve PCNA binding affinity. A ‘linker’ is a covalent tether that connects two distant parts of a peptide sequence to create a bridge and consequently preogranise the peptide backbone into a conformation that is suitable to bind to its target. It has been reported that a linker that affords flexibility in its cyclised structure may be preferable to enable the peptide to adopt its ideal conformation upon binding. This could be provided using the p21μafumRFC peptide by constraining the Arg_143_ and Arg_149_ residues as a linker to cyclise the peptide.

The p21μ-afumRFC secondary structure looks as if it naturally mimics a ring such as that of the bimane structure (Figure 6a). This could be used as a scaffold for a peptidomimetic, which could be improved to be fungal-cell-penetrable, as it has already been shown to not interfere with the 3_10_ helical turn upon binding. The two Arg residues can be replaced to create a linkage that, based on the X-ray crystallography structure, would not interrupt the 3_10_-helical conformation and side chain exposure required for binding to afumPCNA, as these are 3.5 Å distance apart in the naturally forming architecture. The ability to outcompete afumPCNA’s binding in the cell may be achieved via the incorporation of select p21µ-RD2 mutant sequence residues in the p21μ-afumRFC peptide, such as the combination of Tyr_150_ and Phe_151_ aromatic residues, which was shown to be essential in the affinity assay (Table 1).

The bimane peptide (Figure 6b,c) has a unique interaction in which the bimane linker interacts with the C-terminal end of PCNA. This is also achieved in the RFC peptide via the interaction of the proline residue interacting with the C-terminal end of afumPCNA. The key difference in the scaffolds is that the RFC peptide also carried out interactions on the other side of the PIP-box binding domain through the Arg_149_ side chain. This is believed to achieve a more ideal surface packing of the PIP-box onto afumPCNA than previous cyclical peptides have achieved.

Adding a cell-penetrating peptide (CPP) or fungal-specific peptide to the N- or C-terminus of this structure would, based on the X-ray crystallographic structure, not interfere with PIP-box binding and the secondary structure, a problem found for other investigated linkers incorporated into p21 PIP-box peptides. Although high levels of translocation were typically associated with the toxicity of peptides towards fungal cells, SynB, an 18-amino-acid-long peptide, has been found in previous studies of CPPs to be specific for fungal cells with respect to efficiently translocating into the human fungal pathogen *Candida albicans* at concentrations that led to minimal toxicity [39]. Lowered toxicity is vital for experimental studies in assessing the specification of afumPCNA inhibition over fungal cell toxicity. 

## 5. Conclusions

Here, we present the first structure of a p21 PIP-box peptide bound to *A. fumigatus* PCNA, as well as fungal PIP-box candidates, demonstrating the hypothesis that the fungal replication model uses a PIP-box sequence as a method for binding to fungal PCNA.

A high-affinity rational design for a potential cell-permeable peptidomimetic is presented via the combination of a cyclised structure of the p21μafumRFC peptide. Via full cyclisation and the incorporation of select p21µ-RD2 mutant sequence residues, this peptide could be used in the next stages of the drug discovery pipeline as a potential fungal therapeutic. This could be carried out via the addition of a linker to the cyclized secondary structure. Future work will focus on specific fungal cell permeability via the utilisation of the N-terminus of the peptide, which makes no contact with the surface of PCNA and cannot interfere with the helical and cyclized 3_10_ structure.

## Figures and Tables

**Figure 1 jof-09-01098-f001:**
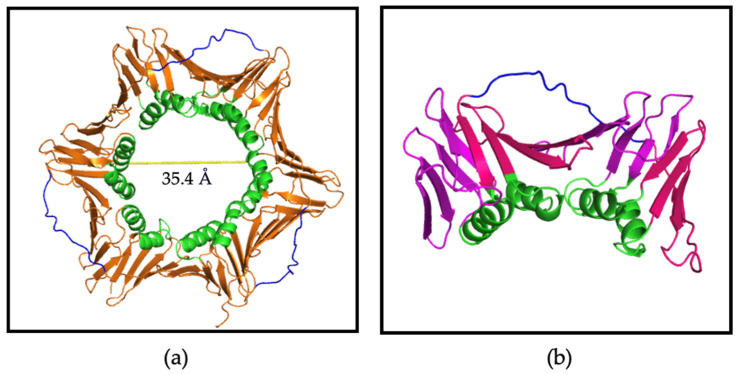
PCNA homotrimer structure (PDB:7KQ0) [18]. (**a**) Trimer: PCNA surface shown in orange, IDCL shown in blue, and positively charged alpha helices shown in green. Central cavity is 35.4 Å wide. (**b**) Monomer: beta sheets shown in magenta and pink, alpha helices shown in green, and IDCL shown in blue. Made using Pymol Version 1.2 [19].

**Figure 2 jof-09-01098-f002:**
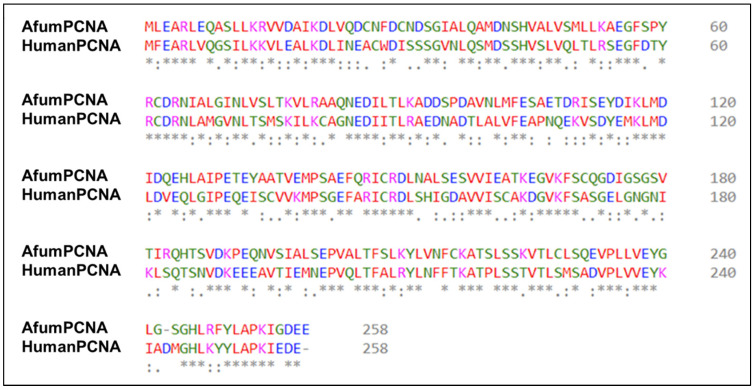
Sequence alignment of afumPCNA (top) (Uniprot: A0A0J5SJF1) vs. Human PCNA (bottom) (Uniprot: P38936). Red residues indicate hydrophobic residues, blue indicates negatively charged residues, pink indicates positively charged residues, and green indicates polar residues. Stars indicate conserved residues. Full stop “.” Indicates residues of shred characteristics. Indicates Semi colon “:” indicates residues of shared characteristics and structure.

**Figure 3 jof-09-01098-f003:**
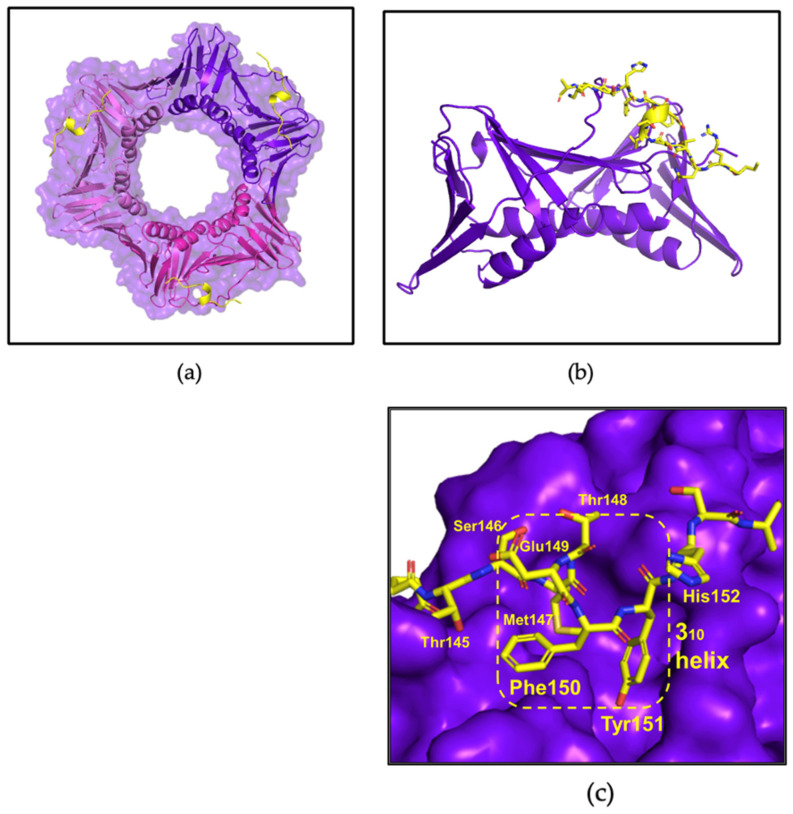
afumPCNA bound with p21µ structure (PDB: 8GJF). (**a**) Trimer: afumPCNA surface shown in purple and p21µ shown in olive green. (**b**) Monomer: afumPCNA shown in purple and p21µ shown in olive green. (**c**) PIP-box binding site: 3_10_ helical structure outlined in yellow and p21µ peptide shown in olive green. Made using Pymol [19]. Crystallographic information can be found in Appendix A.

**Figure 4 jof-09-01098-f004:**
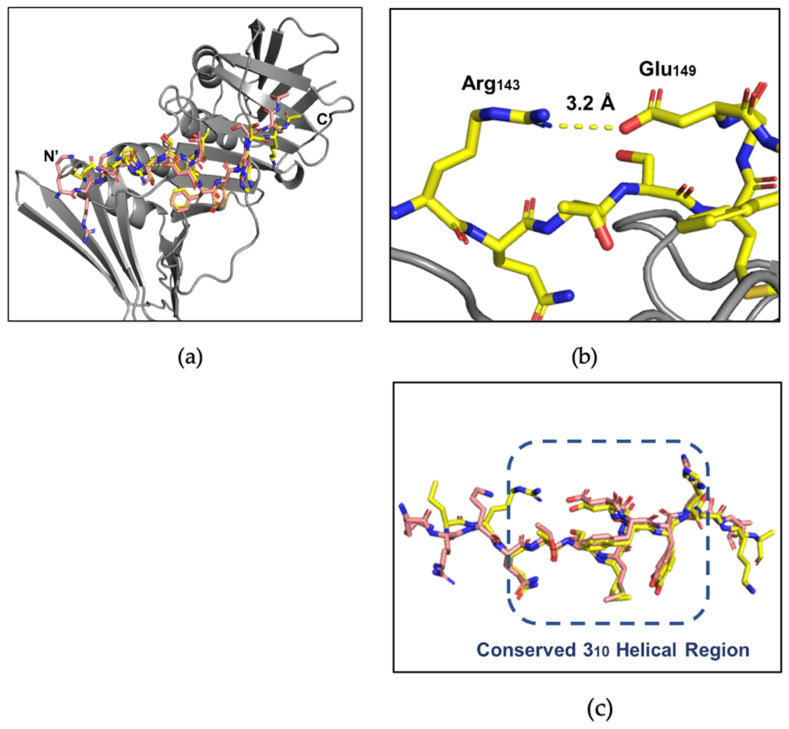
Comparison of the p21µ peptide bound to hPCNA (PDB: 7KQ1) and p21µ peptide bound to afumPCNA (PDB: 8GJF). (**a**) hPCNA shown in grey, p21µ (PDB: 7KQ1) shown in salmon, and p21µ (PDB: 8GJF) shown in yellow. (**b**) Arg_143_ and Glu_149_ salt bridge interaction in p21µ (PDB: 8GJF) structure. (**c**) Conserved 3_10_ helical and PIP-box region of p21µ peptides, residues 144–152: p21µ (PDB: 7KQ1) shown in salmon and p21µ (PDB: 8GJF) shown in yellow (RMSD: 0.939 Å).

**Figure 5 jof-09-01098-f005:**
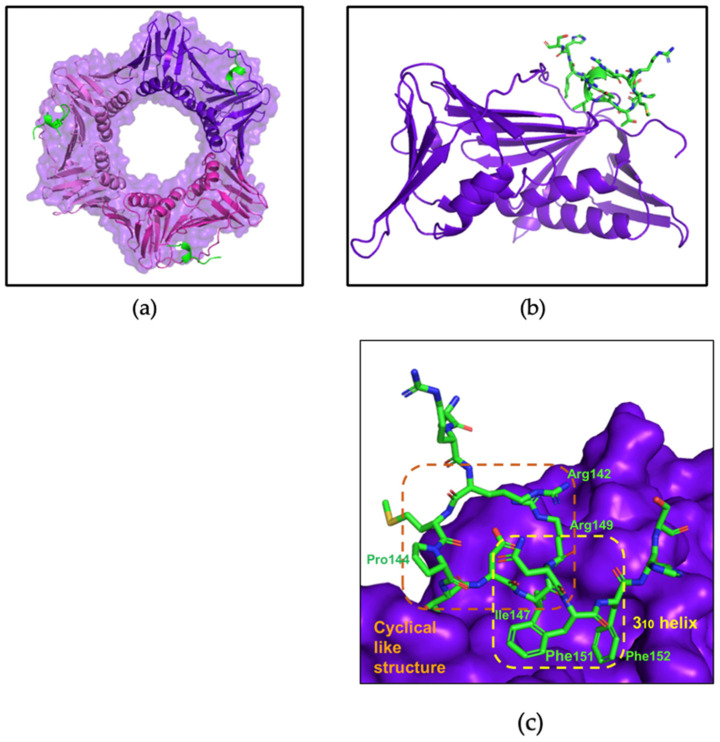
afumPCNA bound with p21µ-afumRFC structure (PDB:8GJ5). (**a**) Trimer: afumPCNA surface shown in purple and p21µ-afumRFC shown in green. (**b**) Monomer: afumPCNA shown in purple and p21µ-afumRFC shown in green. (**c**) PIP-box binding site: 3_10_ helical structure outlined in yellow and cyclical-like secondary structure outlined in orange. p21µ-afumRFC peptide shown in green. Made using PyMOL [19]. Crystallographic information can be found in Appendix A.

**Figure 6 jof-09-01098-f006:**
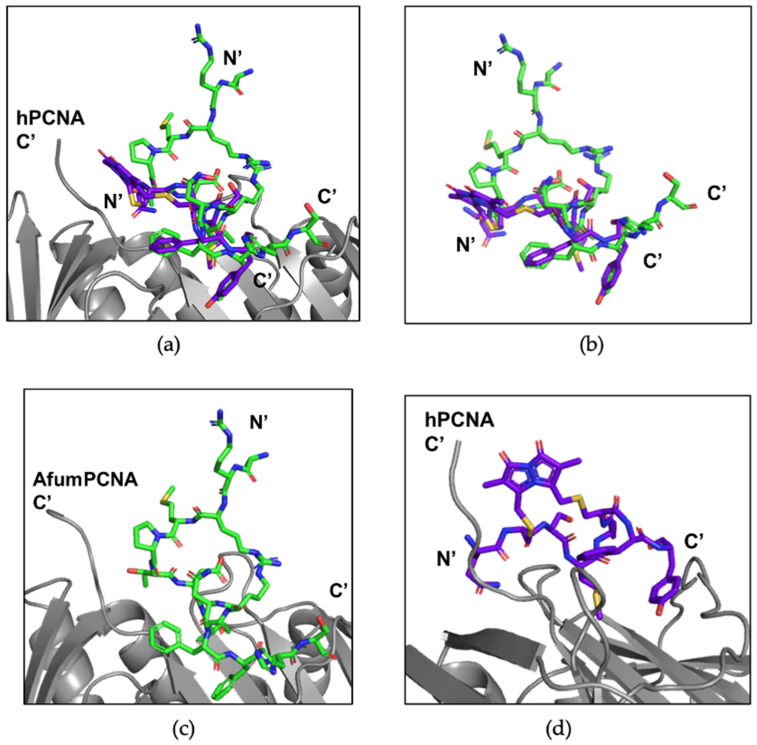
Previous p21-constrained peptides in comparison to the new p21μafumRFC peptide crystal structure. (**a**) p21μafumRFC peptide (green) crystal structure bound to the afumPCNA monomer (grey). p21μafumRFC peptide (green) produced an affinity of 295 ± 6.9 nM to hPCNA. (**b**) The Bimane peptide (purple) docked to the hPCNA monomer (grey), as shown in the computational model, produced an affinity of 570 ± 30 nM to hPCNA [38]. (**c**) p21μafumRFC peptide (green) superimposed over the Bimane peptide (purple) and hPCNA (grey) computational model. (**d**) p21μafumRFC peptide (green) superimposed over the Bimane peptide (purple) computational model.

**Table 1 jof-09-01098-t001:** p21 Peptide SPR data against afumPCNA in comparison to human PCNA binding affinity as shown in Horsfall 2021 [18]. K_D_ is the affinity constant. SD, standard deviation. All peptides are C-terminally amidated. Changes to the p21µ scaffold are indicated in bold. The PIP-box residues are separated from flanking residues with spaces. Conserved PIP positions are underlined. More information can be found in Appendix A.

Name	Sequence	afumPCNA Binding Affinity K_D_ (nM) ± SD (nM)	Human PCNA Binding Affinity K_D_ (nM) ± SD (nM) [18]
p21(139–160)	^139^GRKRR QTSMTDFY HSKRRLIFS^160^	69.7 ± 20.2	4.3 ± 1.3
p21μ	^141^KRR QTSMTDFY HSKR^155^	265.1 ± 5.9	12.3 ± 0.5
p21μ-RD2	^141^KRR QT**RI**T**EYF** HSKR^155^	20.3 ± 6.8	1.1 ± 0.3
p21μ-Q144M	^141^KRR **M**TSMTDFY HSKR^155^	41,400 ± 0.8	1544 ± 159
p21μ-T145K	^141^KRR Q**K**SMTDFY HSKR^155^	10,000 ± 0.7	98 ± 10.8
p21μ-T145D	^141^KRR Q**D**SMTDFY HSKR^155^	4100 ± 0.31	714 ± 30.4
p21μ-S146R	^141^KRR QT**R**MTDFY HSKR^155^	64.4 ± 18.4	4.3 ± 1.3
p21μ-M147L	^141^KRR QTS**L**TDFY HSKR^155^	382 ± 51.0	20.5 ± 1.7
p21μ-M147I	^141^KRR QTS**I**TDFY HSKR^155^	37.1 ± 7.8	11.1 ± 0.3
p21μ-D149E	^141^KRR QTSMT**E**FY HSKR^155^	400.7 ± 45.6	12.7 ± 1.4
p21μ-F150Y	^141^KRR QTSMTD**Y**Y HSKR^155^	75.2 ± 18.9	20.2 ± 0.4
p21μ-Y151F	^141^KRR QTSMTDF**F** HSKR^155^	167.2 ± 19.1	10.6 ± 1.5
p21μ-FY150151YF	^141^KRR QTSMTD**YF** HSKR^155^	96.4 ± 19.6	2.2 ± 0.5

**Table 2 jof-09-01098-t002:** Sequences of candidate fungal protein PIP-boxes in comparison to established human protein PIP-boxes. Sequences that fit the model of an eight-residue section with Q at the beginning, a hydrophobic residue in the middle, and aromatic residues at the end are shown in bold. Residues that are found to be identical between human and fungal PIP-boxes are underlined.

Protein Name	*Homo Sapiens* PIP-Box Sequence	*Aspergillus fumigatus* PIP-Box Sequence Candidates
FEN1	**Q**GR**L**DD**FF**	**Q**SR**L**EG**FF**
RFC1	MD**I**RK**FF**	MPTD**I**RN**FF**
DNA Polymerase POLD3	**Q**VS**I**TG**FF**	**Q**KE**L**SR**F**D
DNA Ligase	**Q**RS**I**MS**FF**	**Q**RVRS**I**AS**FF**

**Table 3 jof-09-01098-t003:** Candidate fungal protein Peptide SPR data against afumPCNA. Peptide p21μafumRFC was also tested against human PCNA. Tested in triplicate. KD is the affinity constant. SD, standard deviation. All peptides are C terminally amidated. Changes to the p21µ scaffold are indicated in bold. More information can be found in Appendix A.

Name	Sequence	afumPCNA Binding Affinity K_D_ (nM) ± K_D_ SD (nM)	Human PCNA Binding Affinity K_D_ (nM) ± K_D_ SD (nM)
p21μafumDNALIG	^141^KRR**QRVRSIASFF**HSKR^157^	458 ± 117.77	-
p21μafumDNAPOL	^141^KRR**QKELSRFDF**HSKR^156^	659.3 ± 105.8	-
p21μafumFEN1	^141^KRR**QSRLEGFF**HSKR^155^	713 ± 56.9	-
p21μafumRFC	^141^KRR**MPTDIRNFF**HSKR^156^	94.84 ± 8.76	295 ± 6.9

## Data Availability

Data are contained within the article or Appendix A. The data presented in this study are available in Appendix A.

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
