# Peer review of "Towards a High-Affinity Peptidomimetic Targeting Proliferating Cell Nuclear Antigen from Aspergillus fumigatus"

_jof, 2023, doi:10.3390/jof9111098_

Round 1

Reviewer 1 Report

Comments and Suggestions for Authors

In the paper entitled "Towards a High Affinity Peptidomimetic Targeting PCNA from Aspergillus fumigatus", the authors present the 1st structure of a p21 PIP-box peptide bound to Aspergillus fumigatus 491 PCNA, as well as some fungal PIP-box candidates, showing somehow that this replication model uses a PIP-box sequence as the method for binding to fungal PCNA.

Considering the rising incidence and mortality of Invasive fungal infections (IFIs), many of them due to A. fumigatus and the urgent need for new antifungals, I believe that the research performed by Vandborg et al. is of great importance.

The paper is well structured, almost perfectly written and could be accepted for publication after some minor changes:

- make sure that Aspergillus fumigatus is always in Italic face (see the title, but also the main text);

- delete the dot ( . ) in the Title;

- insert the abbreviation IFIs (Invasive Fungal Infections), since, in the literature, we mostly find the abbreviation;

- there are some missing (see the references in the text) spaces;

- the Bold face used for some of the peptides should be explained;

- I was wondering if Bibliography could be more up to date, since more than 50% of the references cited are more than 10 years old.

Author Response

“make sure that Aspergillus fumigatus is always in Italic face (see the title, but also the main text)”

We have now corrected this in all instances.

“delete the dot ( . ) in the Title”

We have deleted the dot in the title.

“insert the abbreviation IFIs (Invasive Fungal Infections), since, in the literature, we mostly find the abbreviation”

We have now inserted the abbreviation IFIs.

“the Bold face used for some of the peptides should be explained”

We have taken on the reviewers concerns and altered the following instances that we could find:

2.1 Peptide synthesis sequences shown in bold. (written in text).

Table 1. Changes to p21µ scaffold are indicated in bold. (written in text).

Table 2: Sequences which fit the model of an eight-residue section usually beginning with a Q, the middle a hydrophobic residue, and ending in aromatic residues are in bold. (written in text).

Table 3: Changes to p21µ scaffold are indicated in bold. (written in text).

“I was wondering if Bibliography could be more up to date, since more than 50% of the references cited are more than 10 years old.”

Per this request a number of papers within the last 5 years have been added to the bibliography.

Reviewer 2 Report

Comments and Suggestions for Authors

The authors present a study of synthesized peptides targeting the proliferating cell nuclear antigen (PCNA) of Aspergillus fumigatus.  The goal of the study is to develop peptidomimetic drugs to inhibit DNA replication of A. fumigatus without interfering with DNA replication of human cells. The authors could show that similar mechanisms of p21 like peptides exists like observed in outcompeting human PCNA.

Minor points

Title: p.1: I mention to avoid abbreviations in the title and presenting the full name of PCNA. The species designation Aspergillus fumigatus should be shown in italic letters.

MM, p.4 lines 153, 162, 168, 169, 175 The authors should give information which acid was used to adjust the pH of Tris

Lines 175, 184 The use of % for a solution needs the indication of v/v or w/v for clarification.

Line 184 the concentration of glycine is not described

Line 193 The abbreviation SPR needs to be introduced at the beginning of the chapter

Line 190 “immoblise” means immobilize? 

Line 212:  PEG charges differ from manufacturer in range size of molecules; therefore, indication of manufacturer is important.

Results

p.6, Table 1, last column: SE means SD?

References

Check citations, especially [27] and [28]. Are titles and authors name correct?

Author Response

“Title: p.1: I mention to avoid abbreviations in the title and presenting the full name of PCNA. The species designation Aspergillus fumigatus should be shown in italic letters.”

We have corrected this by avoiding the abbreviation in the title and making Apergillus fumigatus abbreviated in the title.

“p.4 lines 153, 162, 168, 169, 175 The authors should give information which acid was used to adjust the pH of Tris”

The Tris described is pre-adjusted in the tris stock using HCL, this has been corrected in the paper by indication of HCL adjustment by inclusion of “Tris-HCl” denotation.

“Lines 175, 184 The use of % for a solution needs the indication of v/v or w/v for clarification.”

We could not find % at line 174 but at line 184 and therefore corrected it there.

“Line 184 the concentration of glycine is not described”

We have corrected this by adding the concentration.

“Line 193 The abbreviation SPR needs to be introduced at the beginning of the chapter”

This has been corrected by introducing the full name for the abbreviation at the beginning of the chapter.

“Line 190 “immoblise” means immobilize?” 

We have altered this line from British English to American English to bring clarity for the reviewer.

“Line 212:  PEG charges differ from manufacturer in range size of molecules; therefore, indication of manufacturer is important.”

This has been corrected- the supply was inserted with product code.

“p.6, Table 1, last column: SE means SD?”

We have corrected this in the text to the correct term.

“Check citations, especially [27] and [28]. Are titles and authors name correct?”

The names of the citations were correct, but citation format was missing titles. This has been now corrected.

Reviewer 3 Report

Comments and Suggestions for Authors

The manuscript is notably interesting, well-structured, clear, and focused on a contemporary theme within the field of Mycology. The reviewer suggests some minor adjustments before accepting the manuscript for publication in Journal of Fungi.

Title

1. Microorganism names must be presented in italics throughout the manuscript. Please ensure this correction is applied consistently across all sections.

2. Avoid using abbreviations in titles; instead, write out the full term, such as "Proliferating Cell Nuclear Antigen (PCNA)."

Abstract

1. In general, the abstract lacks detailed information concerning the results presented in the manuscript. Therefore, the reviewer strongly recommends augmenting the abstract by including information from the methods section and presenting more comprehensive results to capture the attention of potential readers. Furthermore, it is recommended to craft a more substantial and conclusive final statement.

Introduction

1. Line 43, please, change antibiotic to antifungal.

2. Line 50, please, Change the sentence "fungal inhibitor target" to "fungal target for the development of new antifungals" or a similar phrase.

3. Does Figure 1 contain new results or offer a novel perspective on a known result? If yes, it should be relocated to the Results section. If not, please include a reference that corresponds to the results presented in the figure legend.

Methods and Results

Both sections are well-written, clear, concise, and contain a wealth of new information that is highly relevant to the field of Mycology."

Final suggestions to the authors: The manuscript would benefit from the inclusion of biological assays, examining the potential effects or interference of these new peptidomimetic compounds on A. fumigatus development, encompassing aspects like growth, cell cycle arrest, morphogenesis blockage, among others. Additionally, it is crucial to address the potential toxicity to mammalian cells and the resistance of fungal proteases to cleave these peptidomimetic compounds.

Author Response

“1. Microorganism names must be presented in italics throughout the manuscript. Please ensure this correction is applied consistently across all sections.

  1. Avoid using abbreviations in titles; instead, write out the full term, such as "Proliferating Cell Nuclear Antigen (PCNA)."

We have now made these corrections by italicizing all species and not using abbreviations in the title.

“Abstract

In general, the abstract lacks detailed information concerning the results presented in the manuscript. Therefore, the reviewer strongly recommends augmenting the abstract by including information from the methods section and presenting more comprehensive results to capture the attention of potential readers. Furthermore, it is recommended to craft a more substantial and conclusive final statement.”

We have now attempted these suggestions.  It is important to note that there is only a 200 word limit to the abstract so much more comprehensive information was challenging to add.

“Introduction

Line 43, please, change antibiotic to antifungal.”

We have now corrected this as suggested by the reviewer.

“2. Line 50, please, Change the sentence "fungal inhibitor target" to "fungal target for the development of new antifungals" or a similar phrase.”

We have corrected the text as suggested by the reviewer.

“3. Does Figure 1 contain new results or offer a novel perspective on a known result? If yes, it should be relocated to the Results section. If not, please include a reference that corresponds to the results presented in the figure legend.”

The reference has been inserted into the manuscript to which this structure is first presented.

“Final suggestions to the authors: The manuscript would benefit from the inclusion of biological assays, examining the potential effects or interference of these new peptidomimetic compounds on A. fumigatus development, encompassing aspects like growth, cell cycle arrest, morphogenesis blockage, among others. Additionally, it is crucial to address the potential toxicity to mammalian cells and the resistance of fungal proteases to cleave these peptidomimetic compounds.”

We agree with the reviewer, and these are excellent suggestions.  Unfortunately, this is an extensive amount of new work that we feel is out of the scope of this manuscript.  This manuscript was meant to pursue (and did successfully) a proof of concept for the fungal sliding clamp to identify a PIP-box based inhibitor of reasonable first hit affinity.  The above data will be carried out for a follow up paper.